# Newborn Screening Knowledge, Attitudes and Practices among Obstetrics-Gynecology Residents, Pediatric Residents, and Newborn Screening Nurses in a Tertiary Government Hospital in the Philippines during the COVID-19 Pandemic

**DOI:** 10.3390/ijns9020019

**Published:** 2023-04-01

**Authors:** Patrick Jose D. Padilla, Eileen M. Manalo

**Affiliations:** Department of Obstetrics and Gynecology, Philippine General Hospital, Manila 1000, Philippines

**Keywords:** Newborn Screening, Philippine General Hospital, knowledge, attitudes and practices, RA9288, Newborn Screening Act of 2004

## Abstract

Newborn Screening (NBS) saves babies from mental retardation and death. In the Philippines, it was formally established by law in 2004. Program success requires physicians, nurses, and midwives to educate and motivate parents. The COVID-19 pandemic reduced NBS coverage from 91.6% to 80% between December 2019 and December 2020. This study aimed to (1) determine the knowledge, attitudes, and practices of residents and nurses relative to NBS during the COVID-19 pandemic; and (2) identify possible factors that may have affected NBS services at the Philippine General Hospital during the pandemic. Participants’ demographics were also compared with NBS practices. The study enrolled 189 participants employed during 2020. The results of a self-administered online questionnaire were evaluated. Only 31% of participants scored above a mean passing level for NBS knowledge set by experts. Most participants expressed a favorable attitude towards NBS. Knowledge scores were a significant factor in favorable attitude. Obstetrics-gynecology residents had lower attitude scores than Pediatric residents and NBS Nurses. Prenatal parent education was only practiced by 1/3 of participants. Despite the obstacles of the COVID-19 pandemic, participants appreciated the value of the NBS and were willing to perform specimen collection using safety precautions. Participants identified the need for additional NBS training. The challenges identified provide an avenue for further research with the goal of strengthening NBS, especially during a public health emergency.

## 1. Introduction

Newborn Screening (NBS) is a public health program that aims to identify certain congenital disorders that require immediate intervention to avoid the devastating consequences of mental retardation and death [1,2]. NBS was introduced by Dr. Robert Guthrie in the 1960s in the United States and is currently implemented as part of standard neonatal care in most countries with developed economies [3,4]. NBS started in the Philippines in 1996 [5] and was integrated into the national health delivery system through Republic Act 9288 or the Newborn Screening Act of 2004 [6]. Since its initial offering of six conditions, expanded NBS (ENBS) now includes a panel of twenty-nine conditions [7].

NBS involves absorbing a few drops of blood from a newborn’s heel onto a special filter paper collection card after 24 hours of life. The screening process should be explained to the parent(s) or guardian by a professional health care worker (usually a pediatrician, obstetrician, nurse, or midwife) either during the pregnancy or just before collecting the specimen. Parents (guardians) may refuse the testing. The collection cards are sent to one of seven accredited Newborn Screening Centers (NSCs). Patients with positive screening results are recalled for confirmatory testing and immediately referred to a specialist if needed [7]. Pre-pandemic statistics indicate that newborn coverage in the Philippines was 91.6% in December 2019 [6]. NBS is implemented in 7400+ health facilities (hospitals and birthing homes) throughout the country with a coordinator in every facility [7].

The COVID-19 pandemic has affected NBS operations worldwide, including programs in the Asia Pacific region, North America, Europe, Australia, and other parts of the world. Several pandemic-related NBS challenges were identified including equipment shortages (gloves, kits, reagents, and personal protective equipment); a staffing shortage; inadequate protocols for handling filter cards in the face of known or suspected COVID-19 infections; early discharged patients; and limited courier services [8,9,10,11,12]. In the Philippines, the pandemic affected many facets of the NBS program including courier services, and staff availability, among others. With the onset of pandemic restrictions and lockdowns, healthcare workers caring for pregnant patients needed to be routinely updated on changes in sample collection or other processes, the continued importance of NBS, and their responsibility to assist in patient recall and follow-up [13]. The Newborn Screening Reference Center (NSRC) issued safety protocols to protect newborns, healthcare workers collecting specimens, and NBS laboratory personnel [14]. A reduction in newborn coverage to 80% (a decrease of 12%) was experienced between December 2019 and December 2020 despite the aggressive measures taken by the NSCs and Newborn Screening Facilities (NSFs) [13].

Prenatal care for pregnant women continued during the COVID-19 pandemic, with special precautions and reduced face-to-face doctor–¬patient interactions. Counseling focused on the well-being of the mother and the unborn child, who may need special attention at birth. NBS is a critical topic that should be discussed with pregnant patients as part of routine prenatal care. NBS program knowledge should be an essential part of the armamentarium of all health workers contributing to new mother education, i.e., obstetricians, pediatricians, nurses, and midwives [15]. The American College of Obstetrics and Gynecology recommends including NBS information during prenatal care visits and continuing it postpartum to ensure that newborn caregivers understand its purpose, process, and importance [16,17] and that the optimal time to provide NBS information is during the third trimester of pregnancy, rather than during admission for delivery where there is increased confusion [18].

During the COVID-19 pandemic, it was important to emphasize the need for NBS, how the program was operating despite restrictions, and the protocols in place to ensure the safety of the newborn and healthcare worker collecting blood specimens.

### 1.1. Legislatively Defined NBS Responsibilities

Republic Act 9288 (Newborn Screening Act) became law in 2004, institutionalizing NBS as a healthcare delivery service of the Department of Health [6]. It emphasized the role of health workers in informing parents and guardians to ensure that it would be available and accessible to all newborns.

### 1.2. NBS Information Dissemination

Information on NBS (including ENBS) is widely available, including educational pamphlets at health facilities, posters in hospitals and birthing centers, videos in the waiting areas of some birthing facilities, social media (Facebook, Twitter, etc.), and a NBS program website that provides information to health workers, parents, and the general public [19]. Illustrating the importance of healthcare workers in NBS education, a study conducted at Philippine General Hospital (PGH) showed that 40% of mother-participants identified healthcare workers as their primary source of NBS information, while 43% identified a combination of a healthcare professional and media, society, or relatives [20].

There are few studies [21,22] addressing the status of knowledge, attitudes, and practices of health workers relative to NBS in the Philippines. In order to effectively and appropriately educate prospective parents and guardians about NBS, it is important to identify and address any gaps that might exist in knowledge, attitudes, and practices among healthcare workers. This was especially important during the COVID-19 pandemic since parents were especially apprehensive about birthing issues and since it can serve as a lesson for future disasters.

The study was undertaken primarily to (1) determine the NBS knowledge, attitudes, and practices of OB-GYN, Pedia, and NBS nurses during the COVID-19 pandemic; and (2) to identify possible factors that might have affected the delivery of NBS services at the PGH during the COVID-19 pandemic. A secondary objective was to describe respondent demographics that might be associated with NBS knowledge, attitudes, and practices.

## 2. Materials and Methods

We undertook a study of NBS-related issues in the Philippine General Hospital (PGH), a large tertiary care hospital in Manila, to identify possible gaps in NBS service delivery that might have been occurring during the pandemic. The period of study was 1 January 2021 to 31 December 2021. In this hospital, obstetrics-gynecology residents (OB-GYN), pediatric residents (Pedia), and NBS nurses are the primary healthcare workers who introduce NBS to new parents. A cross-sectional survey using a self-administered online questionnaire was conducted at the PGH. The study and questionnaire were reviewed and approved by the University of the Philippines Manila, Research Ethics Board. The study population included OB-GYN, Pedia, and NBS nurses. NBS nurses were those employed at the time of the study and who voluntarily took charge of NBS education, sample collection, and result release for newborns in the nursery and general services area where babies were roomed-in at the time. E-mail addresses and mobile numbers were obtained from each potential participant’s department. An invitation and link to the questionnaire was sent to each potential participant. Informed consent was integrated into the questionnaire. A total of 189 out of 221 qualified healthcare workers participated in the study with an 86% response rate.

### 2.1. Questionnaire Development

The questionnaire [Appendix A] was patterned after a study by Bansal et al. and revised based on a review of the related literature and an interview with a panel of NBS experts from the NSRC [15]. The questionnaire included demographic data and items designed to obtain information about NBS knowledge, attitudes, and practices during the COVID-19 pandemic. For knowledge scores (dependent variable), age of respondent, sex, profession, and years on the job were the possible factors/predictors considered. For attitude (outcome variable), we considered the association of these same predictors (age of respondent, sex, profession, and years on the job) in addition to NBS knowledge. The expected direction of association was, the higher the level of knowledge, the better is the attitude. Additionally, for the outcome practice (dependent variable), age of respondent, sex, profession, years on the job, NBS knowledge, and attitude score were considered possible predictors. Conceptually, the higher the level of knowledge and attitude scores, the better is the practice.

The questionnaire was validated by a panel of four experts from NSRC. Only questions with items having a Content Validity Index (CVI) of 0.8 (80% of the experts rated an item as quite or highly relevant) were accepted [23]. Questions with a CVI below 0.8 were modified or rejected. The revised questionnaire was pilot-tested on 30 healthcare workers not involved in the project (10 Pedia, 10 OB-GYN, and 10 NBS nurses). Piloting showed a significant correlation, and no further revisions were made. Cronbach’s alpha was used for the questions with a Likert scale response while the split-half correlation coefficient was used for the knowledge assessment questionnaires. The expert panel categorized responses to ‘attitude and practice’ questions as ‘favorable’ or ‘not favorable’ and computed a mean passing level (MPL) for NBS ‘Knowledge’, ‘Attitudes’, and ‘Practices’ questions using the Nedelsky and Angoff methods [24].

Knowledge questions were derived from the NBS/ENBS brochures most recently produced by the NSRC and the NSRC’s memorandum on NBS precautions during the pandemic. Correct answers were verified by the panel of experts. To evaluate attitudes, participants reacted to statements on screening importance, healthcare workers’ roles, the need for training, and the conduct of the screening test during the COVID-19 pandemic. Participants were also asked about their need for further NBS/ENBS training. To assess practices, the participants responded to questions on their frequency of participation in certain practices included in the overall NBS system, such as counseling, the release of patient results, and follow-up/referral for screen-positive newborns. Finally, a series of open-ended questions assessed the effect of the pandemic on NBS service delivery focusing on barriers to effective service delivery and measures that might improve it.

### 2.2. Data Management and Analysis

Data gathered from the participants were encoded in Microsoft Excel. For confidentiality, each response form was assigned a tracking number. Categorical variables (gender, profession, year level of residency, and responses to the ‘attitudes and practices’ assessment questionnaires) were summarized as frequency and percentage. Continuous variables (age, years of practice, and knowledge scores) were summarized by calculating a mean (with standard deviation) and a median (with range). Responses to the open-ended questions were categorized according to theme similarity.

Knowledge questions had specific correct answers, as determined by the panel of experts. Responses for the ‘Attitudes’ and ‘Practices’ questions were coded as either ‘favorable’ or ‘not favorable’ based on the assessment of the expert panel. The scores of the participants for each section (‘Knowledge’, ‘Attitudes’, and ‘Practices’) were then compared based on an MPL set by the expert panel and were categorized accordingly.

Multiple linear regression was used to determine whether age, sex, profession, or years of practice were statistically significantly associated with the scores. Logistic regression analysis was used to determine the associations of various respondent demographics (score ≥ MPL) with a high level of knowledge, favorable attitude, and favorable practice. For attitude, the knowledge score was included as a factor, and for practice, knowledge and attitude scores were included as factors. All estimates were calculated at 95% confidence levels.

Microsoft Excel was used for data management and Stata was used for data analysis.

## 3. Results

### 3.1. Demographics

The mean age of the respondents was 35.3 years (SD = 8.88, median = 38.5, and range = 26–64). Most of the respondents (85.7%) were female. The average years of practice of the nurses was 16.5 years (SD = 8.13, median = 7, and range = 2–28) and the average years of practice as a NBS nurse was 7.29 years (SD = 5.53, median = 7, and range = 1–25). Table 1 shows the demographic characteristics of the participants.

### 3.2. Knowledge

Overall, NBS nurses achieved the highest ‘Knowledge’ scores (17.47/21) followed by Pedia and OB-GYN. On issues of NBS and influences of COVID-19, OBGYN had the highest ‘Knowledge’ scores followed by NBS nurses and Pedia. (Appendix A) Using a ‘Knowledge’ MPL (defined by the NBS Expert Panel) of 72.2%, 85.9% of NBS nurses exceeded the MPL, outperforming both Pedia (81.8%) and OB-GYN (73.5%). (Appendix A).

Among the ‘Knowledge’ questions, the following received less than 75% of correct answers: Q2—need for consent; Q4—ideal time for testing a healthy infant; Q6—number of disorders in the ENBS panel; Q8—appropriate NBS sample collector; and Q16, 18, and 19—filter card handling practices with COVID-19 precautions. For Pedia, the lowest score (54.5%) concerned the practice of packing collection cards for transport from newborns of suspected/confirmed COVID-19 mothers (Q19). For OB-GYN and NBS nurses, the lowest score concerned the need for consent in NBS (Q2). (Appendix A).

### 3.3. Attitudes

A majority of the participants indicated a favorable attitude towards NBS (raw score ≥ MPL): Pedia scored 94.6%, OB-GYN scored 91.8%, and NBS nurses scored 89.4% (Appendix A). Participants’ responses to questions concerning NBS attitudinal variables were also assessed (Appendix A). The majority of the participants agreed that NBS is useful, worth any discomfort a baby might feel, and necessary even if the baby appears healthy at birth. There were, however, mixed responses as to whether a screen-negative newborn could still have a disorder. A lack of comfort in providing counseling to caregivers of screen-positive newborns was also evident from the responses to Q3 (only 28.5% noted strong comfort). There were no significant relationships between the comfort with counseling parents and the year level of a resident or number of years of practice as a NBS nurse. The healthcare workers felt that all play a central role in ensuring that patients benefit from early disorder detection (Q4, Q5, and Q6), and based on the most ‘somewhat agree’ and ‘strongly’ agree’ responses were as follows: Pedia scored 94.7%, NBS nurses scored 92.6%, and OB-GYN scored 88.4%. More than half of the respondents (57.7%) noted a need for more training (Q7). A majority of the respondents, 90.5%, agreed with the statement that they would share NBS information with other physicians and parents (Q9).

Most study participants (81.5%) disagreed with the premise that NBS should have been postponed during the pandemic (Q11), and a similar percentage disagreed that the safety of newborns would be compromised if NBS had been performed during the pandemic (Q12). Most (94.7%) felt that NBS services could have been continued provided that proper personal protective equipment was used by healthcare workers (Q13), and almost all (97.4%) agreed that they would have encouraged parents to have their baby undergo NBS during the COVID-19 pandemic (Q15). Given the uniqueness of the pandemic, a large majority of participants (91.0%) noted a need for additional information about NBS during the pandemic (Q14).

Table 2 lists the areas identified by the Pedia, OB-GYN, and NBS nurses as needing more training and information.

### 3.4. Practices

With the COVID-19 pandemic, there were limited face-to-face explanations on the importance of ENBS. Participants were asked to quantify the frequency of their NBS discussions, including telemedicine. Of particular interest was the number of respondents discussing the rationale for NBS at least 75% of the time. A breakdown of responses (Appendix A) showed that during prenatal care, only 63 of 189 (33.3%) respondents met the 75% requirement. Of these, NBS nurses had the highest practice percentage, 51.8%, followed by Pedia, 29.1%, and OB-GYN, 6.1% (Q1). During admission, only 35 out of 189 (18.5%) met the 75% requirement, with nurses again having the highest percentage, 25.5%, followed by Pedia, 25.4%, and OB-GYN, 2.0%. Postpartum, 66 out of 189 (34.9%) met the 75% requirement, still with NBS nurses being the most compliant, 47.1%, followed by Pedia, 41.82%, and OB-GYN with 6.12% (Q3).

With regard to actively seeking the results of the NBS, only 59 of 189 (31.22%) respondents complied more than 75% of the time. NBS Nurses had the highest practice percentage, 42.4%, followed by Pedia, 40.0%, and OB-GYN, 2.0% (Q4).

Informing parents or guardians of normal results is important so that they will know that the NBS specimen was received at the screening laboratory and that the tests were completed. During the pandemic, only 59 of 189 (31.22%) completed this process more than 75% of the time. The highest percentage reporting normal (screen-negative) results to parents belonged to NBS nurses, 42.35%, followed by Pedia, 20.0%, and OB-GYN, 2.0% (Q5).

Participants who responded that they did not discuss the rationale for NBS, seek results, or inform parents or guardians of normal screening results, were asked who else performed these tasks. Pedia respondents identified Neonatal Intensive Care Unit personnel, NBS nurses, and the NSRC team. OB-GYN respondents identified pediatricians, nurses, midwives, and the NSRC team. On the other hand, NBS nurses identified themselves and the hospital’s NBS coordinator, assigned in every hospital or birthing facility.

For NBS practices, only 58 of 189 respondents (30.7%) had scores above the set MPL of 66.2, with NBS nurses having the highest percentage, 47.1%, followed by Pedia, 29.1%, and OB-GYN, 4.1%. (Appendix A) Respondents were also asked if they knew the agency to contact regarding NBS results. A total of 136 of 189 (72.0%) respondents, 97.7%, mostly the NBS nurses, said they were aware of the agency, followed by Pedia, 80.0%, and OB-GYN, 18.4% (Q6). Regarding knowledge of the appropriate protocol for follow-up, 109 of 189 (57.7%) participants reported knowing the protocol, with the highest percentage being from NBS nurses, 87.1, followed by Pedia, 60.0%), and OB-GYN, 4.1% (Q7).

### 3.5. Association of Respondents’ Demographics with Their Knowledge of and Attitude towards NBS

#### 3.5.1. Knowledge

Multiple linear regression analysis showed that, adjusting for the other variables in the regression equation, such as age and attitude, there were no statistically significant findings with ‘Knowledge’ raw scores.

#### 3.5.2. Attitudes

A multiple linear regression analysis (Table 3) showed that the older a respondent was, the lower was the ‘Attitude’ raw score. For every year older, the score was lower by 0.10 points, which was statistically significant (*p* < 0.001). It is interesting to note that the higher the ‘Knowledge’ score of a respondent, the higher the ‘Attitude’ score. For every increase of 1 point in the ‘Knowledge’ score, there was an increase of 0.33 points, which is statistically significant (*p* < 0.05).

A logistic regression analysis (Table 4) showed that a respondent with favorable knowledge MPL is likely to have a favorable attitude OR of 1.44 (95%; CI: 1.12–1.88).

Another logistic regression model was generated with the ‘Knowledge’ score categorized as low and high based on the MPL. The association of NBS knowledge with the attitude of the respondent was more apparent. A respondent with a favorable Knowledge score higher or equal to the MPL was likely to have a favorable attitude OR: 10.8 (95%; CI: 2.90–40.57). The odds of having a favorable attitude was 11 times more among participants with knowledge scores greater than or equal to the MPL compared to participants with knowledge scores lower than the MPL.

#### 3.5.3. Practice

A multiple linear regression analysis investigating the factors associated with the ‘Practice’ score (Table 5) showed that OB-GYN respondents had lower scores by 2.18 compared to Pedia respondents (*p* < 0.001). The rest of the factors (including high levels of knowledge and a favorable attitude) were not found to have statistically significant associations with the ‘Practice’ score.

‘Practice’ scores were classified as favorable practice (≥MPL score) and unfavorable practice (<MPL score). The only statistically significant variable was profession (Appendix A). OB-GYN respondents were less likely to have favorable ‘Practice’ scores compared to Pedia repondents. NBS nurses were more likely to have favorable ‘Practice’ scores than Pedia respondents but this result was not statistically significant. The rest of the variables also were not statistically significant. A review of the NBS flow of operations showed that the motivation of the parents and the prompt release of results were the steps that were most often not properly or strictly followed.

#### 3.5.4. Challenges and Recommendations

Participants identified various challenges and recommendations for the better implementation of NBS, especially during the COVID-19 pandemic. Table 6 presents the information according to the following categories: education/counseling of parents; education of healthcare workers; improving processes; improving manpower; and improving infrastructure.

## 4. Discussion

The COVID-19 pandemic in 2020–2021 impacted NBS programs globally. The main problems documented were the delays and unreliability of postal services; the unavailability of new NBS kits and other laboratory equipment; the bad sample quality collected outside of the recommended collection time window; staff reductions (due to infection, relocation, or exhaustion); the fact that blood was not taken from children of COVID-19-positive mothers for fear of staff infection; the closure of clinics (their transformation into COVID-19 clinics; the worse communication with field NBS teams; the fear of infection among parents visiting hospitals or the fear of midwives and general practitioners visiting patients at home [11].

Our study is the first to address the knowledge, attitudes, and practices of healthcare workers involved in the Philippine NBS program during the COVID-19 pandemic. While this study was conducted in the context of the pandemic, the questions also addressed the general aspects of NBS. Given the challenges of the pandemic, the results of this study should assist offices in identifying and addressing gaps in the current system. The study also found strong support among respondents for continuing the NBS program as an essential healthcare service (note: the term ‘essential’ here means a program so necessary to the health of the newborn that its operation must be maintained even in the face of a disaster).

While obstetrician-gynecologists provide prenatal education, pediatricians, nurses, and midwives are essential in delivering postpartum care and related education [25]. In many hospitals, nurses are given the lead role in providing NBS information to mothers [26]. At PGH, NBS is led by the NBS Nurses Core Group, and their activities have led to 95–99% newborn coverage [27].

A PGH study found that one group of new mothers was aware of NBS, and had received educative counseling about NBS, but their level of awareness was poor. These mothers reported that the program had not explained NBS in a manner and at a level they could easily understand [20]. This study also found that mothers preferred receiving educational materials, such as pamphlets, with a verbal explanation at an appropriate educational level [20].

Our results show that only 33.3% of healthcare workers discussed the rationale of NBS 76–100% of the time across the various interactions with the parent: 33.3% discussed this during prenatal care, 18.5% discussed this during the admission of the mother, and 34.9% discussed this postpartum. This study emphasizes the need for the inclusion of NBS in the core curriculum of pediatric residency training, obstetric and gynecologic residency training, and nursing students. NBS should also be the subject of continued education seminars and refresher courses, and should be a topic for inclusion in pediatric, obstetric, and nursing conferences. A U.S. study showed that providing pediatric residents with an educational module resulted in significant improved knowledge of NBS [28]. Although not statistically significant, the results showed that the longer a person has been serving in their profession, the higher their raw ‘Knowledge’ score. Statistically significant results showed that the higher the ‘Knowledge’ score, the higher the ‘Attitude’ score. However, a high ‘Knowledge’ score or a favorable ‘Attitude’ score was not found to be statistically significant when associated with the ‘Practice’ score. This suggests the presence of barriers to implementation that fall outside of the healthcare workers’ scope such as issues in processes, infrastructure, and logistics, among others. In fact, evidence supports the need for both provider preparedness and institutional support, such as appropriate infrastructure and culture conducive to giving healthcare providers the opportunity to translate their knowledge to practice [29]. Certain challenges mentioned in this study, such as the lack of manpower for sample collection or tracking paperwork for COVID-19 discordant mothers and babies, must be addressed on an institutional level to promote better NBS operation. Further research may provide solutions to some of these barriers and evaluate further their impact on provider practices. In the meantime, providing an educational module with updated protocols during the pandemic, and reinforcing the need for counseling may improve NBS practice.

There is evidence that healthcare workers’ intention or motivation to carry out specific interventions affect the implementation of these interventions [30]. A positive attitude and the positive intention of the provider may lead to increased provider implementation [30]. Further research should assess healthcare workers’ behavioral intentions with NBS in a pandemic and any effects this might have on program execution.

With the COVID-19 pandemic, and fears of contracting the disease, many workers and parents felt that NBS was not a priority. It was important that safety protocols for both health workers and patients were communicated effectively to allay safety fears. Based on the memorandum released by the NSRC, safety protocols were in place. Our study showed that most respondents agreed that safety protocols contributed positively to NBS activities. Any safety protocols should also be included in training programs for healthcare workers and should also be reinforced during parent counseling.

Our study results showed that there is still a good percentage of study participants who are not comfortable counseling patients about positive screening results. These results are similar to another study among pediatric residents that showed that only 62% were comfortable with counseling, and some were not aware of appropriate follow-up actions for abnormal newborn screening results [15]. A Canadian study showed that a majority of physicians were less likely to discuss NBS than midwives and nurses were [31]. While the reasons for counseling discomfort were not requested in our study, the answer may reside in the fact that 57.7% of all respondents indicated a need for additional training. Hopefully, acknowledging the training need also indicates a willingness to participate in any additional training that might arise as a result of our study.

It appears that many NBS results may not have been properly followed-up during the pandemic since less than one-third of respondents (31.2%) admitted to checking the NBS results over 75% of the time. This was found despite the ready availability of protocols for interpreting and following up on screening results and the fact that some of the screened diseases may require immediate intervention. Flowcharts displayed in NBS collection rooms may serve as a reminder that NBS involves not only specimen collection but also the proper release of results. A flowchart covering the release of abnormal results with corresponding follow-up instructions is also a possible operational aid.

Our study showed that while the majority of respondents knew the appropriate agency to contact regarding results, the knowledgeable respondents were primarily Pedia and NBS nurses, with only 18.4% of OB-GYNs knowing the correct agency. Knowledge of the appropriate protocol for NBS follow-up was highest, 56.7%, in NBS nurses. While care of the newborn may not seem to fall under the umbrella of obstetric care, it is still important for OB-GYNs to be aware of NBS follow-up protocols in order to respond to questions during postpartum follow-ups and to ensure that screening test results are known.

The various challenges and recommendations revealed in this study provide insights into possible NBS program and policy improvements, especially regarding the pandemic. The logistical and operational concerns identified require resolution. Further studies may be required to investigate specific concerns and solutions. The positive attitude of survey respondents towards maintaining the NBS program should be complemented by training and refresher courses emphasizing patient and worker safety during the COVID-19 pandemic. Because continuing NBS education is specified in NBS law, additional focus on safety during the pandemic may be added to operational protocols.

Our study has identified a number of gaps in knowledge that translate to inadequate or improper service delivery, particularly during the COVID-19 pandemic. While our experiences were in a large tertiary care hospital, it is likely that they can be applied to many hospitals and birthing centers throughout the country. Information gathered from this study will be forwarded to the Department of Health and the NSRC to help in their information campaigns and the improvement of the Newborn Screening Program.

## 5. Conclusions

Despite the obstacles of the COVID-19 pandemic, all healthcare workers in the study acknowledged the value of NBS and were willing to perform NBS with appropriate health safety precautions. The identification of gaps and strengths in NBS knowledge and attitudes and practices among healthcare workers provides opportunities for improvement in the national NBS program. The education of healthcare workers, patients and guardians play a critical role in the acceptance of NBS at birth. Educational information provided during prenatal visits, knowledge levels of associated healthcare workers, and delivery of additional screening information postpartum all contribute to the success of NBS. Recognition of the importance of NBS by OB-GYN, Pedia and nurses during the COVID-19 pandemic and beyond was demonstrated by our study. The study also highlighted the necessity to continuously educate healthcare workers on all aspects of NBS so that they can in turn educate parents.

## Figures and Tables

**Table 1 IJNS-09-00019-t001:** Demographic characteristics of participants.

VARIABLE	FREQUENCY (%)
Mean age in years (SD) Gender	35.3 (8.88)
Male	25 (13.20%)
Female	162 (85.70%)
Prefer not to say	2 (1.10%)
Profession	
Pediatric Resident	55 (29.10%)
Obstetrics and Gynecology Resident	49 (25.90%)
Newborn Screening Nurse	85 (45.00%)
Years of Medical Residency	
Pediatric Resident	
1st Year	18 (32.70%)
2ndYear	19 (34.50%)
3rd Year	18 (32.70%)
Obstetrics and Gynecology Resident	
1st Year	12 (24.50%)
2ndYear	16 (32.70%)
3rd Year	15 (30.60%)
4th Year	6 (12.20%)
Years of Practice as Nurse	
1–8 years	16 (18.81%)
9–16 years	28 (32.94%)
17–24 years	24 (28.22%)
25–32 years	15 (17.65%)
33–37 years	2 (2.36%)
Years of Practice as Newborn Screening Nurse	
1–4 years	27 (31.77%)
5–8 years	22 (25.88%)
9–12 years	24 (28.23%)
>13 years	12 (14.1%)

**Table 2 IJNS-09-00019-t002:** NBS topics subjects identified by study participants as requiring additional training.

Actual process of NBS Sample collection (P,O,N) Handling of filter cards from babies from COVID-19-positive mothers (N) Logistics (where offered) (O) Storage and processing (P,O,N) Releasing of results (P,O,N) Interpretation of results (N,O)
Health education/Counseling (P,O,N)
Confirmatory testing Appropriate confirmatory testing for positive screen results (P,N) Policies for referral (P,N)
Knowledge about Expanded NBS Diseases included in ENBS (P,O,N) Limitations of specific screening tests (P) Policies on Prematurity (P,N)
Expanded NBS and COVID-19 Policies on precautions (P,O,N)

Abbreviations: P = pediatric residents; O = OB-GYN residents; N = NBS nurses; NBS = newborn screening; ENBS = expanded newborn screening.

**Table 3 IJNS-09-00019-t003:** Results of the multiple linear regression analysis; outcome: attitude score.

Attitude Score	Adjusted Coefficient	Std Error	*p*-Value	95% Confidence Interval	
				Lower	Upper
Age, years	−0.10	0.03	0.00	−0.15	−0.05
Profession (compared to Pedia)					
OB-GYN	−0.67	0.35	0.06	−1.36	0.02
NBS nurses	0.03	0.43	0.94	−0.81	0.88
Years in profession (years of service in one’s profession)	0.06	0.04	0.12	0.02	0.14
Knowledge	0.33	0.06	0.00	0.20	0.46

Abbreviations: NBS = Newborn Screening; OB/GYN = Obstetrics-gynecology residents; Pedia = pediatric residents.

**Table 4 IJNS-09-00019-t004:** Results of logistic regression analysis; outcome: favorable attitude (score ≥ MPL).

Attitude Score	Adjusted Odds Ratio	Std Error	*p*-Value	95% Confidence Interva	
				Lower	Upper
Age, years	0.86	0.05	0.01	0.76	0.96
Profession (compared to Pedia)					
OB-GYN	0.76	0.62	0.73	0.15	3.76
NBS nurses	3.77	4.82	0.30	0.31	46.36
Years in profession (years of service in one’s profession)	1.07	0.07	0.33	0.93	1.23
Knowledge	1.44	0.19	0.01	1.12	1.88

Abbreviations: NBS = Newborn Screening; OB-GYN = Obstetrics-gynecology residents; Pedia = pediatric residents.

**Table 5 IJNS-09-00019-t005:** Results of the multiple linear regression, Outcome: Practice Score.

Practice Score	Adjusted Coefficient	Std Error	*p*-Value	95% Confidence Interval	
				Lower	Upper
Age, years	−0.03	0.03	0.19	−0.08	0.02
Profession (compared to Pedia residents)					
OB-GYN residents	−2.18	0.33	0.00	−2.84	−1.52
Nurses	0.59	0.42	0.16	−0.23	1.40
Years in profession (years of service in one’s profession)	0.05	0.04	0.18	−0.02	0.13
Knowledge—mean passing level	0.45	0.34	0.18	−0.21	1.11
Attitude—mean passing level	−0.24	0.48	0.62	−1.18	0.70

**Table 6 IJNS-09-00019-t006:** Challenges and recommendations for improving NBS service delivery during the COVID-19 pandemic from pediatric residents, OB-GYN residents and Newborn Screening nurses.

Challenges		Recommendations	
		Continue	Strengthen/Enhance
**Education and counseling of parents/guardians**	Prompt education and counseling of parents (O) with special attention given to teen mothers (N); Difficulty in explaining NBS to a COVID-19-positive mother in a limited face-to-face setting, especially when the mother is separated from the baby (P); Myths and misconceptions of the value of NBS during the COVID-19 pandemic (N);	Awareness campaign through distribution of educational materials (P, O, and N); Online materials explaining NBS/ENBS to parents (P); Webinars and infomercial videos (N);	Improve the health-seeking behavior of parents, in general (P), by introducing NBS as part of the Newborn Package (free), as early as during initial and follow-up prenatal checkups (P, O, and N); Enhance participation of OB-GYNs and pediatricians to encourage their patients to submit their newborn to be screened (N); Integrate telemedicine for discussions on NBS (P); Consider for inclusion in the materials for parents: facts and myths on the value of NBS during COVID-19 pandemic (N); Provide pamphlets in the vernacular (N);
**Education of healthcare workers**	Lack of education of HCWs on NBS/ENBS and COVID-19 protocols (P and O); Face-to-face training compromised by COVID-19 (P); Need for skills in motivating parents to agree to NBS during the COVID-19 pandemic (P);	Orientation/refresher seminars and workshops for pediatric and OB-GYN residents to include heel prick (P, O, and N); Lectures on additional protocols during the pandemic (P and N);	Hold mandatory training workshops/meetings/conferences for nurses, pediatricians and obstetricians on NBS/ENBS to include skills on motivating parents and guardians (P); Include the topic of NBS during the pandemic in lectures (P and N);
**Protecting the safety of patients and healthcare workers from babies of COVID-19-positive mothers**	Safety of the patient during return to the facility for repeat testing or confirmatory testing (P and N); Safety of the healthcare personnel during sample collection, while handling filter cards (P, O, and N), and while releasing results (P)	Provision of precautionary measures, i.e., PPE for staff (P, O, and N); Proper segregation of specimen for positive and negative specimens (for COVID-19) (N);	Establish clear guidelines for ensuring safety during the sample collection, i.e., completing filter card information outside the COVID-19 ward (N);
**Improving manpower**	Lack of trained staff to perform sample collection (P, O, and N)	Regular orientations of the staff (P, O, and N);	Train more personnel to perform NBS per area (P, O, and N); Provide a dedicated NBS screener including the follow-up and release of results (N)
**Improving processes**	Separation of COVID-19-positive mothers from newborns leading to tracking problems (P); Timeliness of sample collection due to the added precautions of COVID-19 (O); Availability of NBS filter cards (P); Prompt release of results to ensure prompt recall for repeat testing or confirmatory testing (P and N); Delayed confirmatory testing due to quarantine; financial need for transportation to laboratory (N); Stoppage of courier services during ECQ (N); dedicated place for sample collection of premature babies coming back at day 28 (N);	Algorithms on how to do confirmatory testing if positive and stressing that this should not be delayed despite the pandemic (P); A more streamlined process of contacting and notifying both the caregiver and institution of the positive results of the NBS (P); More aggressive implementation of specimen collection immediately after 24 h of life (P);	Establish policy for the electronic release of results to parents (N and P); Explain the value of knowledge in securing a NBS result prior to discharge (O and P); Give an appointment schedule for parents to avoid crowding (O); Provide regular collection even on weekends (O); Designate a Newborn Screening area (O) Provide funds for confirmatory testing for all conditions (N); [note: program covers confirmatory testing for metabolic conditions] Provide alternative modes for sample submission to the laboratory in times of ECQ (N);
**Improving infrastructure**	-	Allowing more centers to be capable of handling COVID-19-suspected or COVID-19-positive mothers (P); Better ventilated rooms for sample collection (N);	Secure online database for access to results and a hotline for online consultations (P); Coordination with local health centers or barangay health stations for follow-up and repeat collection (N); Dedicated office for the operations, to handle all components (information, testing, release of results, follow up for repeat testing, and recall for confirmatory testing) (N); Provision of assistance to families of babies with positive results incapable of accessing quality healthcare (N);

Abbreviations: P = pediatric residents; O = OB-GYN residents; N = NBS nurses; COVID-19 = coronavirus 2019; ECQ = enhanced community quarantine; ENBS = Expanded Newborn Screening; HCW = healthcare workers; NBS = Newborn Screening; OPD = out-patient department; PPE = personal protective equipment.

## Data Availability

All data generated or analyzed during this study are included in this published article.

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
