# Peer review of "Newborn Screening Knowledge, Attitudes and Practices among Obstetrics-Gynecology Residents, Pediatric Residents, and Newborn Screening Nurses in a Tertiary Government Hospital in the Philippines during the COVID-19 Pandemic"

_2409-515X, 2023, doi:10.3390/ijns9020019_

Round 1

Reviewer 1 Report

This article on knowledge, attitude and practice of newborn screening (NBS) in the Philippines and the impact of the COVID 19 pandemic addresses an interesting and important topic. The results show, among other things, that attitudes and knowledge are higher among experienced health workers. Nevertheless, participants pointed out the need for additional NBS training, especially additional information on NBS during the pandemic. In my opinion, this interesting article can be improved by focusing on some findings and including other points in the discussion.

Introduction:

Information about the responsibilities in NBS in the Philippines or this hospital should be included in the introduction line 44-48 as this is important for the results. It appears, that OB-Gyn (please explain abbreviation in the abstract) are not responsible for NBS and therefore don’t practice it.

Line 39: Reference 5 is from 2006 and should be replaced (i.e., Padilla CD et al 2015…)

Line 55: one reference (8-12) is enough for the different parts of NEWSTEP.

Line 79-103: is very detailed and can be omitted in my opinion

Materials and Methods:

Language of the questionnaire? English? Some questions seem to be addressed to parents and not to professionals (i.e., 9., 15.?)

Please add the reference for the Nedelsky and Angoff method (line 156) and the version of the programs used (line 188).

Results:

Table 1: In my opinion, years of practice as nurse are not relevant for the results (remove from table 1), and the last three categories for practice as NBS nurse can be summarized as the frequency is very low. On the other hand, age seems to be a significant factor and the distribution should be included in this table.

In my opinion, chapter 3.4 “Practice” seems to be an organizational issue of NBS in the hospital and so it might not be useful to compare practice scores between the professions. I would suggest deleting this part of the results (3.4, 3.5.3 and table 6) and discussion (line 419-425, 454-469) or at least excluding the OB-GYN.

Sex should not be included in the logistic regression analysis as there are only 2 participants in the category “others” and there might be a large selection bias. The results of table 5 could suffice in the text as they provide little additional information to table 4.

Maybe table 7 can be removed to the supplement

Discussion:

In my opinion, the discussion should focus on knowledge and attitude and there influencing factors in the different professions. Prenatal information about NBS is not the subject of this paper and not addressed in the results and therefore should not be discussed. I would suggest deleting line 373-379, 394-396, 397-401. On the other hand, the comparison with other articles about NBS during the COVID-19 pandemic (there are at least 5 papers listed in Pubmed with the search newborn screening AND Covid-19 in 2021-2022) is completely missing.

Conclusion:

Line 485-486??

Adapt line 492-494

References:

31 and 32 are not cited in the text!!

References have to be adapted if some parts of the discussion are excluded.

Please make sure, that the citation of websites is always in the same style.

Author Response

Dear Reviewer,

Thank you very much for your comments to our submitted manuscript.  Your comments have been most helpful  in improving our manuscript. I am also attaching the pdf document with the point-by-point responses.

Respectfully,

Patrick Jose Padilla

RESPONSES

Introduction: 

  1. Information about the responsibilities in NBS in the Philippines or this hospital should be included in the introduction line 44-48 as this is important for the results. It appears, that OB-Gyn (please explain abbreviation in the abstract) are not responsible for NBS and therefore don’t practice it. 

RESPONSE: OB-GYN was spelled out -  Obstetrics-gynecology

A sentence has been added to clarify the role of OB-GYN and further clarify the screening process.

NBS involves absorbing a few drops of blood from a newborn’s heel onto a special filter paper collection card after 24 hours of life.  The screening process should be explained to the parent(s) or guardian by a professional health care worker (usually a pediatrician, obstetricians, nurse, or midwife) either during the pregnancy or just before collecting the specimen.

  1. Line 39: Reference 5 is from 2006 and should be replaced (i.e., Padilla CD et al 2015…)

RESPONSE: We have replaced Ref. 5 with a more up-to-date review of the current status of NBS.

Therrell, B.L.; Padilla, C.D.; Loeber, J.G.; Khneisser, I.; Saadallah, A.; Borrajo, G.J.; Adams J. Current status of newborn screening worldwide: 2015. Semin Perinatol. 2015, 39, 171-187. doi: 10.1053/j.semperi.2015.03.002

  1. Line 55: one reference (8-12) is enough for the different parts of NEWSTEP.

RESPONSE: reduced to 1 reference

  1. Line 79-103: is very detailed and can be omitted in my opinion

RESPONSE: deleted as suggested

  1. Materials and Methods:

Language of the questionnaire? English? Some questions seem to be addressed to parents and not to professionals (i.e., 9., 15.?)

RESPONSE: The questionnaire was in English and seemed to be well understood by the participants.

The questions were addressed to professionals; no parents were involved.  Q9 refers to the knowledge of the professional on timing of availability of results.

  1. Please add the reference for the Nedelsky and Angoff method (line 156) and the version of the programs

RESPONSE; Reference added

Thompson, N. Angoff and Nedelsky Standard Setting Procedures. Assessment Systems Corporation, 23 July 2022. Available online: https;//assess.com/nedelsky-method/ (accessed on 14 March 2023).

Results:

  1. Table 1: In my opinion, years of practice as nurse are not relevant for the results (remove from table 1), and the last three categories for practice as NBS nurse can be summarized as the frequency is very low. On the other hand, age seems to be a significant factor and the distribution should be included in this table. 

RESPONSE: No changes made. We were interested in knowing/showing whether length of practice as a nurse or newborn screening nurse affected NBS program perceptions and practices. NBS nurses were those employed at the time of the study who voluntarily took charge of NBS education, sample collection and result release for newborns in the nursery and general services area where babies were roomed at the time.

  1. In my opinion, chapter 3.4 “Practice” seems to be an organizational issue of NBS in the hospital and so it might not be useful to compare practice scores between the professions. I would suggest deleting this part of the results (3.4, 3.5.3 and table 6) and discussion (line 419-425, 454-469) or at least excluding the OB-GYN. 

RESPONSE - We have not made any changes here.  One of the more important points of this study was demonstrating the degree to which the various professionals (and therefore professions) understood their role in NBS. In particular it was important to demonstrate that there was a role for each of the professions in executing the NBS program. The data demonstrated the lack of importance given to NBS by OB/GYNs in particular and the need to build up their understanding of the program and its importance.

  1. Sex should not be included in the logistic regression analysis as there are only 2 participants in the category “others” and there might be a large selection bias. The results of table 5 could suffice in the text as they provide little additional information to table 4.

RESPONSE- This has been addressed. Sex was removed as a variable and this section has been re-written with new calculations without the variable ‘sex’..

  1. Maybe table 7 can be removed to the supplement

RESPONSE: We prefer to  retain this table because it represents a summary of improvements. It has been shortened. Some parts were merged.

Discussion:

  1. In my opinion, the discussion should focus on knowledge and attitude and there influencing factors in the different professions. Prenatal information about NBS is not the subject of this paper and not addressed in the results and therefore should not be discussed. I would suggest deleting line 373-379, 394-396, 397-401.

RESPONSE. We revised the paragraph. The introduction of NBS during prenatal visits has been shown to be critical for acceptance of NBS upon the delivery of birth. This is important especially for programs in developing countries as awareness/education must start before/during pregnancy.

  1. On the other hand, the comparison with other articles about NBS during the COVID-19 pandemic (there are at least 5 papers listed in Pubmed with the search newborn screening AND Covid-19 in 2021-2022) is completely missing. 

RESPONSE.  Additional references have been added.

Conclusion:

  1. Line 485-486??

Response This was deleted

  1. Adapt line 492-494

RESPONSE: The paragraph has been modified.

References:

  1. 31 and 32 are not cited in the text!!

RESPONSE: removed as suggested.

  1. References have to be adapted if some parts of the discussion are excluded. 

RESPONSE: References have been adjusted.

  1. Please make sure, that the citation of websites is always in the same style.

RESPONSE: DONE

Reviewer 2 Report

Dear author(s),

Thank you for submitting your work and allowing me the opportunity to provide some comments and feedback. The manuscript entitled ‘‘Newborn Screening Knowledge, Attitudes and Practices Among Obstetrics-Gynecology Residents, Pediatric Residents, and Newborn Screening Nurses in a Tertiary Government Hospital in the Philippines During the COVID-19 Pandemic’’  Overall, I believe the work to have importance and I think it will be of use to other researchers. Please find my comments and feedback below with reference to Lines.

Lines 35-37: Authors should cite the origins of neonatal screening rather than citing their own sources (References 1-3).

Lines 39-41: the law should be cited as the source.

6 and 7 references seem to be the same. their access times are different. Access URL or reports should be given to access information.

(6. Padilla, C.D. (Newborn Screening Reference Center, University of the Philippines, Manila, Manila, Philip-554 pines). Personal communication. 11 October 2021. 555

7. Padilla, C.D. (Newborn Screening Reference Center, University of the Philippines, Manila, Manila, Philip-556 pines). Personal communication. 5 June 2021.)

Lines 120-124 (We undertook a study of NBS-related issues in the Philippine General Hospital (PGH), a large tertiary care hospital in Manila, to identify possible gaps in NBS service delivery that might be occurring during the pandemic. In this hospital, Obstetrician-Gynecology Residents (OB-GYN), Pediatric Residents (Pedia), and NBS Nurses are the primary healthcare workers introducing NBS to new parents). This information should be given after line 134 in the material section.

Lines 113-119- is this opinion of the researchers or past research suggests it too? Just need a small clarification here. The main contribution is not made so clear, perhaps through more discussion of literature and identification of gaps in our understanding would help to clarify this. The answers to the following questions must be added to the beginning paragraph. What was there in the few studies that addressed the topic? Why did you need this study? What was the main gap? The answers to these questions should be clearly clarification.

2. Materials and Methods

It is worth defining the research design. The date range of the research should be clearly clarification. Information about the universe and sample of the research should be presented clearly.

Line 149- (Only questions with items having a Content Validity Index (CVI) of 0.8 (80% of the experts rated an  item as quite or highly relevant) were accepted. Questions with a CVI below 0.8 were modified or rejected.) It would be nice to add a reference for this information.

The description of the tool should have highlighted the dependent variable and key predictors, in line with the purpose of the paper. The scoring of the questions should be clearly stated.

Line 168- You modified the word “mangement” as “management

3. Results

Line 191- (A total of 189 out of 221 qualified healthcare workers participated in the study with an 86% response rate). This information should be taken into the material and method section.

Line 192- (Table 1 shows the demographic characteristics of the participants.) This sentence can be taken at the end of the paragraph.

S2- Participants’ level of Knowledge on NBS and Knowledge on NBS and COVID-  Statistical comparisons can be made according to occupational groups in the tables. Statistical method should be chosen according to whether it shows a normal distribution or not.

Lines 281-286 - Information here should be removed. This information may be misleading as there is no statistically significant result.

Lines 289-291 It should be removed as well.

Lines 299-301 It should be removed as well.

Similar statements should be removed elsewhere. Similar information should be removed in the discussion section

Could there be a problem in the Table 4? What is the % explanatory power of the model?  This information is important and should be given below the tables.

4. Discussion

Lines 359-379-  This information can be included in the introduction. In addition, it should be enhanced to include comparisons with other studies on this subject.

Author Response

Dear Reviewer 2,

Thank you very much for your comments to our submitted paper. Your comments have been most helpful in improving our manuscript. The attached document provides the responses to your comments.

Respectfully,

Patrick Jose D. Padilla

Round 2

Reviewer 1 Report

The paper is much better now, and some points have been addressed  in this version of the article. But I still think that  the chapter "practice"  should be edited as suggested and the influence of the COVID 19 pandemie in other countries on  NBS should be included in the discussion. I think this is for the editors to decide.

Smaller points:

Line 149: the language should be added: A cross sectional survey using a self-administered online questionnaire in English was conducted at the PGH

I still think, that the questionnaire in the supplement might be at least in parts for parents, i.e., Attitude assessment question 9: I will share information regarding newborn screening to other parents and / or peers, Question 15: I will encourage other parents to have their child undergo newborn screening…. This questionnaire should be exchanged.

Again: in Table 1 age of the respondents should be added, as this seems to be an influencing factor. Categories with very small numbers could be summarized (i.e., Years of practice as a newborn nurse > 13 years n = 12 instead splitting up in 3 groups

Author Response

The paper is much better now, and some points have been addressed  in this version of the article. But I still think that  the chapter "practice"  should be edited as suggested and the influence of the COVID 19 pandemic in other countries on  NBS should be included in the discussion. I think this is for the editors to decide.

Response.

As recommended, the discussion has been updated to show the effect of the COVID 19 pandemic in the other countries, with particular reference to the paper entitled ‘Global impact of COVID-19 on newborn screening programmes’ by Koracin et al. This has been added as a first paragraph of the discussion.

     The COVID-19 pandemic in 2020-2021 impacted NBS programs globally. The main problems documented were: delays and unreliability of postal services; unavailability of new NBS kits and other laboratory equipment; bad sample quality collected outside of the recommended collection time window; staff reduction (due to infection, relocation or exhaustion), blood was not taken in children from COVID-19-positive mothers for fear of staff infection; closure of clinics (transformation to COVID-19 clinics; worse communication with field NBS teams;  fear of infection among parents visiting hospitals or fear of midwives and general practitioners visiting patients at home [11].

With regards the chapter on 'practice', we appeal to the editor to allow us to keep the information on the OB-Gyne. For our hospital, it is not an organizational issue.  Hospitals in developing countries have to strategize how to implement successfully.  The recommendations of this paper are now being addressed by the Newborn Screening Reference Center. To improve NBS implementation, we recruit nurses from both Obstetrics and Pediatrics wards to join the core of Newborn Screening Core group.  This is voluntary. There is continuous recruitment.  Just last Tuesday, there was an orientation for 39 volunteer nurses (employed by the hospital and does extra work of education, collection etc after duty) and half were from obstetrics.

For the obstetricians, the orientation and reorientation is still being scheduled.

This paper will be distributed to all the hospitals and we want to highlight the equal roles of all sectors. 

Smaller points:

Line 149: the language should be added: A cross sectional survey using a self-administered online questionnaire in English was conducted at the PGH

Response: Edited as recommended

I still think, that the questionnaire in the supplement might be at least in parts for parents, i.e., Attitude assessment question 9: I will share information regarding newborn screening to other parents and / or peers, Question 15: I will encourage other parents to have their child undergo newborn screening…. This questionnaire should be exchanged.

Response: We thank you for your suggestion. Since the study is already completed, we will consider this for a future study involving parents.

For this study, all respondents were health professionals and we believe there was no confusion in the interpretation of the question.

Again: in Table 1 age of the respondents should be added, as this seems to be an influencing factor.

Response: Edited as recommended.

Categories with very small numbers could be summarized (i.e., Years of practice as a newborn nurse > 13 years n = 12 instead splitting up in 3 group

Edited as recommended.

Reviewer 2 Report

Dear authors,

in my opinion you considered all the comments and suggestions, clearly improving the article, and I think it is in good conditions to be published.

Author Response

Thank you very much for recommending the publication of the paper.

Most appreciated.

Patrick Padilla